# Breakdown of bulk-projected isotropy in surface electronic states of topological Kondo insulator SmB$_6$(001)

Yoshiyuki Ohtsubo [1,2,3] ✉, Toru Nakaya[3], Takuto Nakamura [2,3],
Patrick Le Fèvre[4], François Bertran [4], Fumitoshi Iga [5] &
Shin-Ichi Kimura [2,3,6] ✉

The topology and spin-orbital polarization of two-dimensional (2D) surface electronic states have been extensively studied in this decade. One major interest in them is their close relationship with the parities of the bulk (3D) electronic states. In this context, the surface is often regarded as a simple truncation of the bulk crystal. Here we show breakdown of the bulk-related in-plane rotation symmetry in the topological surface states (TSSs) of the Kondo insulator SmB$_6$. Angle-resolved photoelectron spectroscopy (ARPES) performed on the vicinal SmB$_6$(001)-$p$(2 × 2) surface showed that TSSs are anisotropic and that the Fermi contour lacks the fourfold rotation symmetry maintained in the bulk. This result emphasizes the important role of the surface atomic structure even in TSSs. Moreover, it suggests that the engineering of surface atomic structure could provide a new pathway to tailor various properties among TSSs, such as anisotropic surface conductivity, nesting of surface Fermi contours, or the number and position of van Hove singularities in 2D reciprocal space.

The close correspondence between symmetry operations in 3D bulk bands and the topological character of the surface states lying on the edge of a crystal has been one of the central topics of solid state physics in this decade[1]. Such topological surface states (TSSs) are attractive not only for basic science but also for spintronic applications due to their momentum-dependent spin–orbital polarization[2,3]. Since TSSs originate from the topological order of bulk states, some qualitative characteristics of TSSs, such as the metallic band dispersion across the bulk bandgap, are robust against any non-magnetic external perturbations as long as the bulk states remain unchanged. This is a strong point on the one hand because of the expected contamination-tolerant working of devices, for example. On the other hand, the role of the surface atomic structure in TSSs is regarded as rather unimportant because

of the robustness. Some theoretical and experimental studies reported modification of TSSs by surface treatments, such as surface oxidization[4], chemical or photochemical ageing[5,6], and a small interlayer rotation forming a moiré modulation on the surface[7]. However, such previous works were focused on how to perturb the pristine TSS. Actually, the unit cells of the surface lattice in these works were always the same length of basis vectors as the ideally truncated bulk crystal, with the symmetry operations obtained by simple projection of bulk 3D space group. Some studies focused on topological electronic states localized at 1D edges or hinges, but they still suppose the similar simple projection from 3D to 1D systems[8,9]. Although some drastic electronic phenomena, such as 1D charge-density-wave formation on 2D semiconductor surfaces[10], obtained by forming a surface atomic structure independent of the

[1]National Institutes for Quantum Science and Technology, Sendai 980-8579, Japan. [2]Graduate School of Frontier Biosciences, Osaka University, Suita 565-0871, Japan. [3]Department of Physics, Graduate School of Science, Osaka University, Toyonaka 560-0043, Japan. [4]Synchrotron SOLEIL, L'Orme des Merisiers, Départemental 128, F-91190 Saint-Aubin, France. [5]Graduate School of Science and Engineering, Ibaraki University, Mito 310-8512, Japan. [6]Institute for Molecular Science, Okazaki 444-8585, Japan. ✉e-mail: y_oh@qst.go.jp; kimura.shin-ichi.fbs@osaka-u.ac.jp

substrate, are already known, the role of the surface superstructures in TSSs has not been studied in detail yet, to the best of our knowledge.

Samarium hexaboride ($SmB_6$) is a long-known Kondo insulator, in which a bulk bandgap opens at low temperature because of the Kondo effect[11]. It is the first material proposed as a candidate for topological Kondo insulators (TKIs), and it hosts a metallic TSS coexisting with strong electron correlation[12,13] and was recently confirmed as a TKI by angle-resolved photoelectron spectroscopy (ARPES) experiments after a long debate[14–18], as summarized in ref. 19. The (001) surface of $SmB_6$ is also known as a valuable example of a well-defined surface superlattice among TIs. Although cleavage provides multiple surface terminations[20], in situ surface preparation, typically by the cycling of Ar ion sputtering and annealing, results in an uniform surface superstructure such as (1 × 2) and (2 × 2)[14,22], making the $SmB_6$(001) surface a good template to study the role of the surface atomic structure in TSSs. The remaining barrier for such research is that the obtained $SmB_6$(001) surfaces have two or more equivalent domains coexisting with nearly the same area. For example, double-domain (1 × 2) and (2 × 1) surfaces could provide apparent fourfold rotation symmetry of the TSS by overlapping of two twofold domains rotated 90° to each other, as observed on the Si(001) surface[23].

In this work, we show the TSS on the uniform and semi-single-domain $SmB_6$(001)-$p(2 × 2)$ surface prepared in situ. One surface domain is dominantly grown by using a vicinal (001) substrate, confirmed by low-energy electron diffraction (LEED). The dispersion and orbital–angular-momentum (OAM) polarization of the $SmB_6$(001) TSS is observed by ARPES without ambiguity from the multidomain overlap for the first time. The ARPES data show that the TSS is anisotropic and that the Fermi contour (FC) lacks the fourfold rotation symmetry that is maintained in bulk. This result emphasizes the important role of the surface atomic structure even in the TSS.

## Results

### Vicinal (001) surface of $SmB_6$

As depicted in Fig. 1a, the bulk-truncated (001) surface of $SmB_6$ has a fourfold rotation symmetry. After cycles of Ar ion sputtering and annealing (see "Methods" for details), the vicinal $SmB_6$(001) surface exhibited the LEED patterns shown in Fig. 1b. In contrast to most of the earlier studies reporting (2 × 1) surface periodicity[14,21], the (1/2 1/2) fractional spot indicated the $p(2 × 2)$ surface structure at the primary electron energy ($E_p$) of 65 eV. Moreover, other fractional spots at $E_p = 45$ eV, $(1 − 1/2)$ and $(1/2 − 1)$, show different intensities from each other, although they should be identical if the surface has fourfold rotation and time-inversion symmetries. Note that different diffraction spots become bright at $E_p = 45$ and 65 eV simply because to sweep LEED $E_p$ corresponds to sweep the size of the Ewald sphere in reciprocal space. The asymmetric intensities between $(1 − 1/2)$ and $(1/2 − 1)$ are quantitatively shown by the LEED line profiles in Fig. 1c. This shows that the obtained surface has an anisotropic atomic structure without fourfold rotation symmetry. The line profile also shows that this surface hosts the mirror planes [100] and [010], since the spots $(1 ± 1/2)$ have nearly the same heights. Note that the height difference between $(1 − 1/2)$ and $(1/2 − 1)$ does not directly reflect the area ratios of surface domains; the obtained surface periodicity is $p(2 × 2)$, and thus both $(1 − 1/2)$ and $(1/2 − 1)$ spots could appear even from the perfect single-domain surface.

Figure 1d shows the B 1$s$ core level and $Sm^{2+}$ 4$f$ valence bands obtained from angle-integrated photoelectron spectra of $SmB_6$(001)-$p(2 × 2)$. Different from the cleaved $SmB_6$(001) cases[17], we found no clear "surface" term which appears at different energies from the main peaks. This would occurs because the heating process during the surface preparation removes the metastable Sm and B atoms which are the origin of the "surface" terms. The B 1$s$ peak has a tailed shape with respect to higher kinetic energies, suggesting that some boron atoms are in a different surrounding environment from the bulk case, but the difference is not as drastic as in the "boron-terminated" case of the cleaved surfaces[17,20]. This suggests that small displacements of the

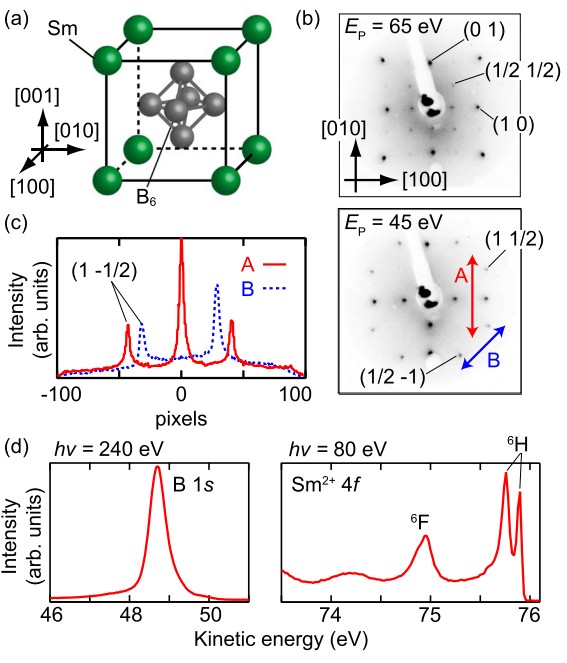

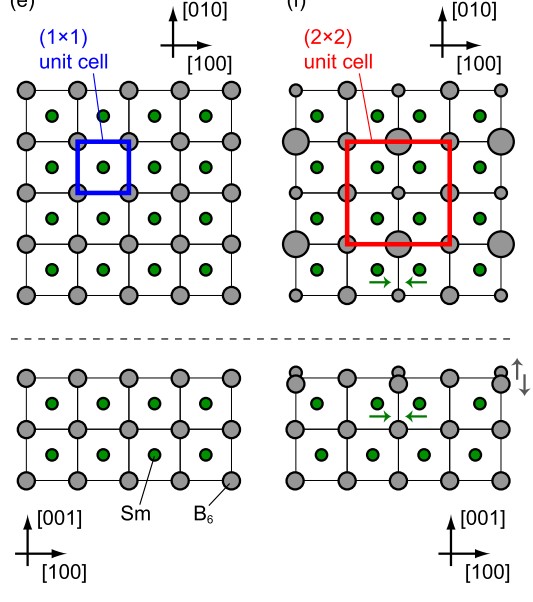

**Fig. 1 | Surface preparation of the vicinal $SmB_6$(001) surface. a** Crystal structure of $SmB_6$. **b** LEED patterns of the cleaned $SmB_6$(001) surface taken at 100 K. The distortions in the patterns are due to the flat microchannel plate in the LEED electron optics. Two-headed arrows indicate the lines along which the profiles in (**c**) were taken. **c** LEED line profiles taken from the pattern shown in (**b**). **d** Angle-integrated photoelectron spectra taken at 13 K. $^6$F and $^6$H are the $Sm^{2+}$ 4$f^5$ final states after photoexcitation. **e** A surface atomic structure of $SmB_6$(001) without any surface superstructure. **f** The same as (**e**) but with hypothetical $p(2 × 2)$ surface reconstruction. Arrows guide the displacements of Sm and $B_6$ near the surface.

surface boron atoms are the origin of the $p(2 \times 2)$ superlattice. While some cleaved surfaces of SmB$_6$(001) were reported to be inhomogeneous with various termination atoms[17,20,24], we found no such signature from the current surface. Further discussion on this point is shown in Supplementary Note 1.

Based on these results, we propose a possible model of $p(2 \times 2)$ surface superstructure, as illustrated in Fig. 1e and f. In this model, no surface defects nor adatoms are supposed, and the $p(2 \times 2)$ superlattice is formed only by displacements of B$_6$ and Sm atoms at the topmost two atomic layers. Such picture agrees with the less drastic change of surrounding conditions of the surface atoms, which was suggested by the core-level spectra explained above. In addition, the number of neighboring atoms of the displaced Sm is the same as those in bulk, in contrast to the reduction for the topmost B$_6$. It rationalizes the core-level spectra showing the "surface" term only for boron peaks. Note that this model is just a possible example to satisfy what were observed by LEED and core-level spectra. Further experiments are required to determine the surface atomic structure accurately, such as dynamical LEED analysis[25] or Weissenberg reflection high-energy electron diffraction[26], while the $p(2 \times 2)$ surface unit cell without fourfold rotation symmetry is enough for the following discussion in this article.

### TSS of the vicinal SmB$_6$(001)-$p(2 \times 2)$ surface

Figure 2a shows the FCs around the Fermi level ($E_F$) measured with circularly polarized photons at 35 eV. The spectra obtained by using both right- and left-handed polarizations are summed to avoid the anisotropic intensity distribution due to the circular dichroism (CD). Figure 2b shows a schematic drawing of the obtained FCs S1, S2, and S3. The oval enclosing $\bar{X}$ (S1) has a similar shape to those observed in the earlier studies[15–17,19,21]. The stark contrast to the earlier studies is that the FC around the other $\bar{X}$ point, S1′ depicted in Fig. 2b, is quite weak, although they should be identical, if the surface has a fourfold rotation symmetry. To highlight this difference, the other $\bar{X}$ point is named as $\bar{Y}$ (Fig. 2a). One possibility to explain this anisotropy is the ARPES experimental geometry. However, as illustrated in Fig. 2c, the $\pm k_{y//[010]}$ orientations are identical to each other, even when the photoelectron incidence orientation is taken into account. Moreover, S1′ becomes as evident as S1 near the edge of the crystal surface where the vicinal miscut is expected to be different from that at the center of the polished surface, indicating that the difference between S1 and S1′ is from the surface domains (ARPES data are shown in SM). Together with the anisotropic LEED patterns (Fig. 1b, c), the faint S1′ can reasonably be assigned to the minor-area domains and to the fact that the SmB$_6$(001)-$p(2 \times 2)$ surface lacks the fourfold rotation symmetry with the S1 FC only around one $\bar{X}$ point.

The other FCs enclosing $\bar{\Gamma}$, S2 and S3, are also clearly observed, in contrast to the blurred FCs in earlier works[16]. This is due to the uniform single-domain surface preparation over a wide area of the sample. The shape of S2 is similar to that observed on the cleaved (001) surface. While that state was claimed as the umklapp of S1 by (1 × 2) surface periodicity at first[17], this assignment has not yet reached a consensus, as the following discussion made a counterargument on this assignment[19]. On the other hand, as shown in Fig. 2a and b, the shapes of S1 and S2 observed here are clearly different, indicating that S2 is another, independent FC.

All the three FCs observed here lacks the fourfold symmetry, as shown in Fig. 2. Such breakdown of the bulk-related symmetry in the TSS is observed for the first time among TIs, to the best of our knowledge. It should be derived from the anisotropic surrounding conditions of Sm atoms near the surface, since the bulk states around the Kondo gap is mainly derived from Sm 4$f$ and 5$d$ orbitals. Such condition can be naturally satisfied by assuming $p(2 \times 2)$ surface superstructure; for example, minor displacements of Sm below the topmost surfaces as depicted in Fig. 1f. On the other hand, no folding

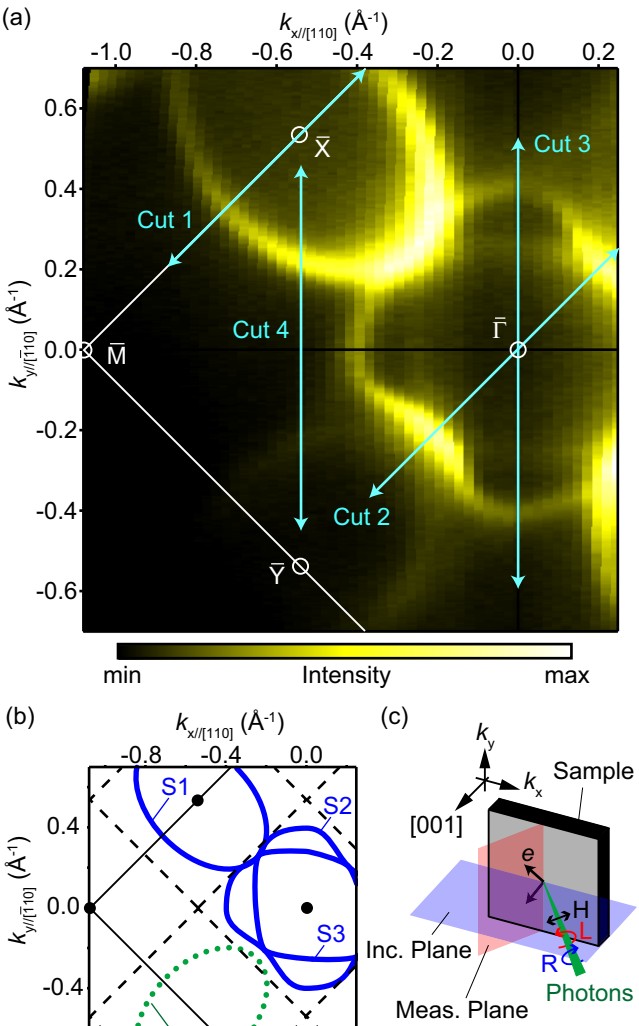

**Fig. 2 | Fermi contours (FCs) obtained by ARPES. a** ARPES FCs taken with circularly polarized photons ($h\nu = 35$ eV) at 13 K. The ARPES intensities from the left- and right-handed polarizations are summed up to show all the states without any influence of circular dichroism (CD). The photon-incident plane is ($\bar{1}10$). The arrows indicate the positions where the $E$-$k$ dispersions shown in Figs. 3, 4, and Supplementary Fig. 1 were taken. **b** Schematic drawing of the observed FCs together with the border of surface Brillouin zones; solid lines for bulk-truncated (1 × 1) and dashed for (2 × 2). **c** Experimental geometry and definition of the in-plane wavevectors $k_x$ and $k_y$. $k_x$ and $k_y$ are always in the photon-incident and photoelectron detection planes, respectively. In this work, two cases, the incident planes of (010) and ($\bar{1}10$) were measured. The relationship between $k_{x,y}$ and the crystal orientation is shown by subindexes such as $k_{y//[\bar{1}10]}$ and $k_{y//[010]}$.

of FCs at the SBZ boundary was observed, while it is rather common case among surface states with superstructure, as discussed in Supplementary Note 2.

Figure 3 shows the band dispersions near $E_F$. The almost localized Sm 4$f$ band lies slightly below $E_F$ (~20 meV) and the Sm 5$d$ bands disperses in 150−50 meV, reflecting the itinerant character. Around the crossing point between them, the bands bend, indicating the $c$-$f$ hybridization induced by the Kondo effect. Above the Sm 4$f$ band, the metallic bands continuously disperses between $E_F$ and Sm 4$f$ in Cuts 1−3. Their Fermi wavevectors ($k_F$) are consistent with the FCs observed in Fig. 2. Note that distinguishing S1, S2 and S3 in Fig. 3b is difficult because they overlap with each other at $E_F$ along this orientation. While the surface bands corresponding to S2 and S3 are not very clear from the 2D intensity plot in Fig. 3c, this is because of the too strong Sm 4$f$ band and the momentum distribution curve (MDC) at $E_F$ shows distinct

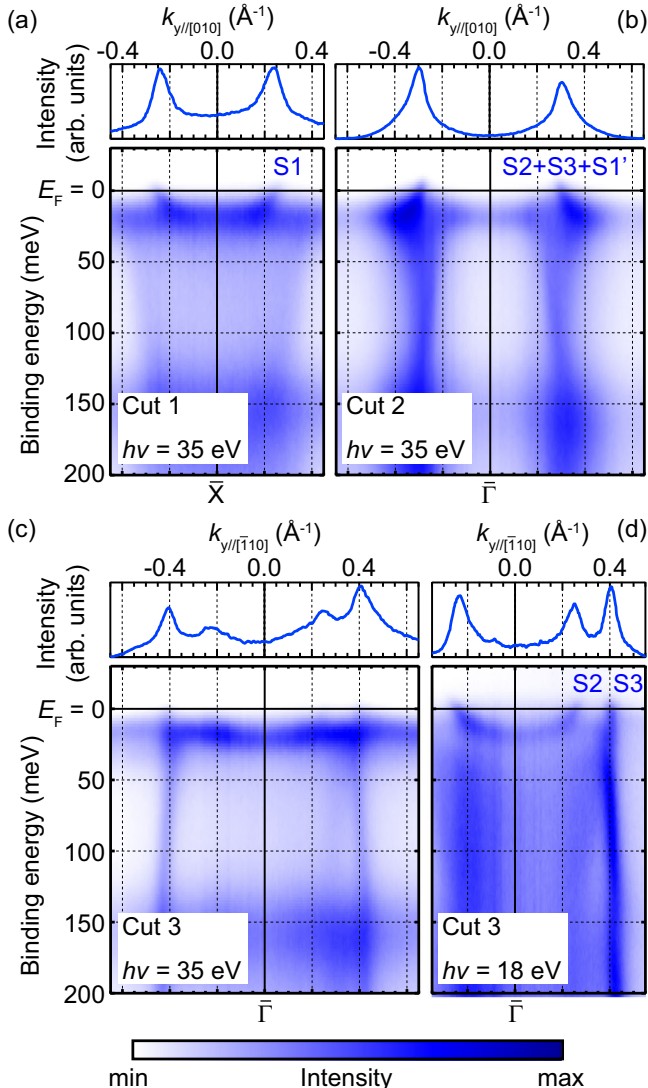

**Fig. 3 | Band dispersions of SmB$_6$(001)-$p(2 \times 2)$.** ARPES intensity plots with the same condition as in Fig. 2 together with the momentum distribution curves cut at $E_F$, measured along the (**a**) X̄-M̄ (Cut 1), (**b**) Γ̄-X̄ (Cut 2), and (**c, d**) Γ̄-M̄ (Cut 3) orientations.

peaks corresponding to them. Moreover, Fig. 3d shows the dispersion of S2 and S3 close to $E_F$ without such difficulty. This is thanks to the photon energy (18 eV) for Fig. 3d, where the photoemission cross-section for Sm 4$f$ becomes small. The $k_F$ values for S2 and S3 obtained in Fig. 3c and d show no difference depending on the photon energy, indicating that their 2D nature arises from the surface. For further confirmation, we also checked the lack of 3D dispersion for S2 and S3 by measuring a series of MDCs at $E_F$ with different photon energies (see Supplementary Fig. 2 in SM).

**OAM polarization of the TSS**

One of the most prominent characteristics of TSSs is the helical spin and OAM polarization, which is always perpendicular to the in-plane wavevector[1]. Figure 4 shows a CD-ARPES map corresponding to the region shown in Fig. 3. The CD of ARPES reflects the OAM polarizations projected onto the photon-incident orientation. As illustrated in Fig. 2c, CD in the current experimental geometry corresponds to OAM polarization along the in-plane and normal to $k_y$, or out-of-plane orientation. Figure 4a shows that the oval-shaped S1 around X̄ has helical OAM polarization consistent with the earlier CD-ARPES[15] and

spin-resolved ARPES[16] results. Figure 4b and c show that the surface bands S2 and S3 also have OAM polarizations whose sign inverts from $+k_y$ to $-k_y$, obeying the time-inversion symmetry. This behavior, helical OAM polarizations with time-reversal symmetry, is consistent with what is expected for TSSs[1,19].

Figure 4d illustrates the OAM polarizations of each FC, assuming that they are parallel to the helicities of the incident-photon polarizations: parallel (anti-parallel) to the photon-incidence vector for right- (left-) handed polarization. According to this, the helicities of the OAM polarizations for S1 (counterclockwise) are opposite to those for S2 and S3 (clockwise). Such behavior is not expected from the analytical calculation with the winding number[27]. However, note that the sign of the CD-ARPES signal is not suitable information to be directly compared with such calculation of the initial states. First, it is easily affected by the photoelectron excitation process[28]. Second, the calculation does not include the surface atomic structure obtained in this work. Finally, TSSs were recently revealed to often contain multiple wave-function components with different orbital and spin polarizations even at single $(E, k)$ point; thus, the polarization of photoelectrons is not always the direct information of the initial state[29,30]. Therefore, further datasets, especially spin-resolved ARPES with multiple incident-photon energies and polarizations, are required to conclude the winding number of the TSS.

## Discussion

We have revealed that the TSS of SmB$_6$(001)-$p(2 \times 2)$ is highly anisotropic. From this anisotropy, the other electronic properties, such as the electron conductivity and net spin polarization via the surface currents, are also suggested to be anisotropic on SmB$_6$(001)-$p(2 \times 2)$, as far as they are derived from the TSS. Note that none of these properties are topologically protected. Therefore, examining the rotation asymmetry from the vicinal SmB$_6$(001) sample with in situ surface preparation would be an interesting new pathway to distinguish the origin and dimensionality (2D or 3D) of unconventional electronic phenomena observed in SmB$_6$, such as the thermodynamic properties[31] and quantum oscillations[32–37]. If one could apply the surface preparation method similar to ours to vicinal SmB$_6$(001) sample and then insert it to a magnet equipment for de Haas van Alphen oscillation measurements without breaking the vacuum environment, such experimental setup would provide conclusive evidence about the role of surface on the quantum oscillation.

Since the folding of FCs by surface superstructure changes the apparent shape of FCs drastically, one could have a question about its role on topological order. In most cases including SmB$_6$(001)-$p(2 \times 2)$, surface superstructure causes no change the energetic order of bulk bands nor magnetic order. Therefore, there shouldn't be any change in topological order before and after the formation of surface superstructure. However, it would be helpful to discuss how to determine the topological order with the surface superstructure, since it is often determined by counting the number of FCs to be odd or even, around surface time-reversal invariant momenta (TRIM); Γ̄, X̄, Ȳ, and M̄ for SmB$_6$(001)[18,19,38]. For this counting, we propose to double check the number of FCs both by using (1 × 1) bulk-truncated SBZ as well as that with folding according to the surface super-structure. Circled numbers in Fig. 4d and e are examples of this method applied to SmB$_6$(001)-$p(2 \times 2)$, showing 3 FCs, odd number consistent with the non-trivial topological order[18,19]. Note that S1 at the bottom side of Fig. 4d is identical to the top one because of the translational symmetry, and thus it should not be included to this FC counting. This method emphasizes the importance to prepare the single-domain surface to discuss the topological order of unknown material; for example, surface Dirac cones doubled by two none-quivalent surface terminations, as observed in ref. 39, could derive false counting of FCs.

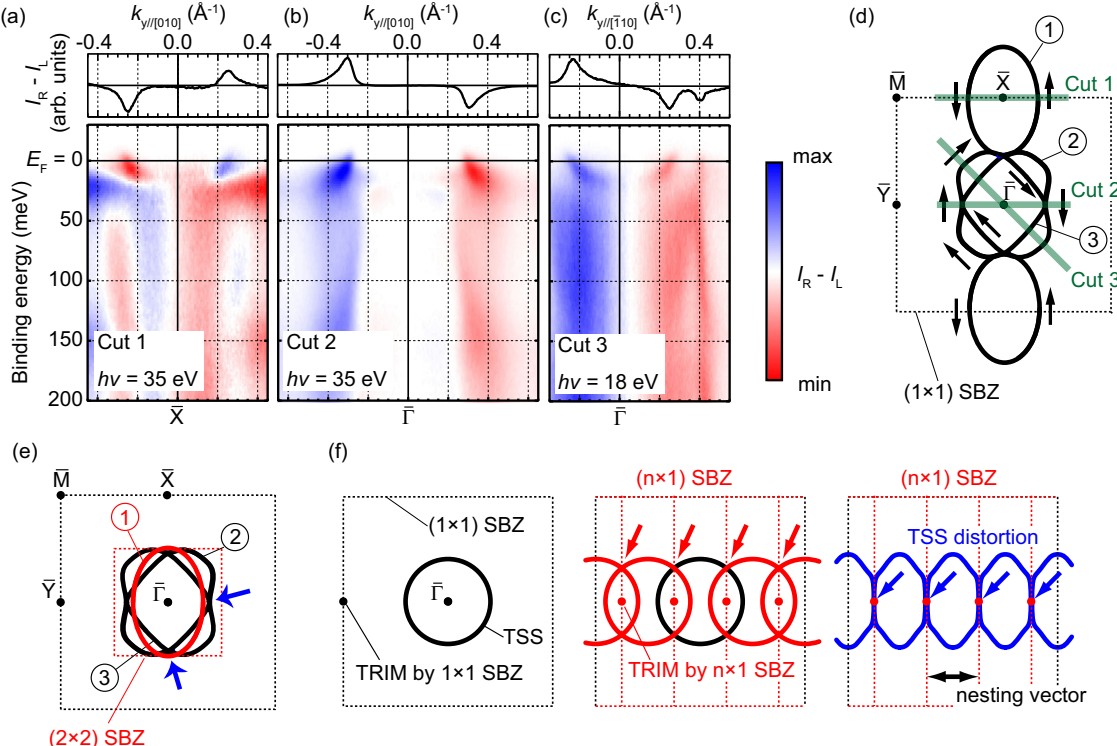

**Fig. 4 | CD-ARPES band dispersions. a–c** CD-ARPES plots taken at 13 K. The measurement geometries are the same as those in Figs. 2 and 3. **d** Schematic drawing of the OAM polarizations corresponding to surface FCs, assuming that they are parallel to the helicities of the incident-photon polarizations: parallel (anti-parallel) to the photon-incidence vector for right- (left-) handed polarization. Fat lines indicate the cut orientations shown in (**a–c**). Circled digits indicate number of surface FCs in SBZ. **e** Surface FCs folded by (2 × 2) SBZ boundary. Arrows guide the possible singularity points (see main text for details). **f** Schematic drawings of TSS with folding and distortion from the anisotropic surface superstructure.

The folding of TSS by surface superstructure could play further role on the unconventional electronic phenomena of TSS. It duplicates FCs formed by TSS, making the additional crossing points with van-Hove singularity (VHS, arrows in Fig. 4f). Such VHS formation is discussed theoretically based on a moiré modulation to expect enhanced topological superconductivity[40]. Moreover, this work exhibited that the surface superstructure distorts surface FCs at the same time. It suggests that engineering of surface superstructure could tailor the number and position of VHS in the reciprocal space as well as the other parameters of FCs, such as nesting vector dominating surface density-wave formation[41]. Actually, the observed FC on SmB$_6$(001)-$p$(2 × 2) does exhibit triple degenerate points of TSSs as guided by arrows in Fig. 4e. On the other hand, the reconstruction of TSS in Kondo gap observed here is the first case, to the best of our knowledge. In most cases, surface superstructure are regarded to play a major role in larger energy scale typically in 0.1–1 eV, orders of magnitude larger than those of Kondo gap (a few tens of meV). The VHSs formed by surface superstructure possibly also enhance the electron correlation effect expected to TSS of TKI, such as the emergence of heavy surface states and non-Hermitian exceptional points[42,43]. From these perspectives, detailed electron behavior in smaller energy and temperature scale, in orders of few meV and Kelvins, respectively, of SmB$_6$(001)-$p$(2 × 2) would be an encouraging test peace to examine the role of the surface superstructure on TKI.

## Methods
### Sample preparation
Single crystalline SmB$_6$ was grown by the floating-zone method by using an image furnace with four xenon lamps[21,44]. The sample cut along the (001) plane was mechanically polished in air with a small angle offset toward [010] until a mirror-like shiny surface was obtained

with only a few scratches when observed under an optical microscope (multiple ×10 magnification). The miscut angle is estimated to be ~1° from the side-view images obtained by the optical microscope. The polished sample was moved to ultra-high-vacuum (UHV) chambers and cleaned in situ by repeated cycles of Ar ion sputtering (1 keV) and annealing up to 1400 ± 50 K. The cleaned surface was transferred to the measurement chambers for LEED or ARPES without breaking the UHV.

### LEED measurements
The LEED measurements were performed by using a conventional electron optics (OCI, Model BDL800IR) at beamline 7U of UVSOR-III. The sample was mounted on a homemade 6-axis goniometer maintained at 100 K. The electron incident angle was adjusted to be parallel to the surface normal ([001]) by using the rotation angles of the goniometers.

### ARPES experimental setup
The ARPES measurements were performed with synchrotron radiation at the CASSIOPÉE beamline of synchrotron SOLEIL. The photon energies used in these measurements ranged from 18 to 240 eV. The polarization of incident photons is depicted in Fig. 2c, including linear polarization with the electric field lying in the incident plane (H) and circular polarizations (R and L). The photoelectron kinetic energy at $E_F$ and the overall energy resolution of the ARPES setup (~15 meV) were calibrated using the Fermi edge of the photoelectron spectra from Ta foils attached to the sample.

## Data availability
The datasets generated and/or analyzed during this study are available from the corresponding authors upon reasonable request.

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

## Acknowledgements

We acknowledge W. Breton and F. Deschamps for their support during the experiments on the CASSIOPÉE beamline at the SOLEIL synchrotron. We also thank S. Ideta and K. Tanaka for their support for LEED measurements at UVSOR with proposal No. 20-784. The ARPES experiments were performed under SOLEIL proposal No. 20191629. This work was also supported by JSPS KAKENHI (Grants Nos. JP20H04453, JP19H01830, and JP20K03859).

## Author contributions

Y.O. conducted the ARPES experiments with assistance from P.L.F. and F.B., Y.O., T. Nakaya, and T.Nakamura performed the LEED experiments. F.I. grew the single-crystal samples. Y.O. and S.-i.K. wrote the text and were responsible for the overall direction of the research project. All authors contributed to the scientific planning and discussions.

## Competing interests

The authors declare no competing interests.
