## [Peer Review File · Nature Communications]

REVIEWERS' COMMENTS

Reviewer #2 (Remarks to the Author):

The authors have responded substantively and satisfactorily to all the many comments that I made in a report on the version of the manuscript that was submitted to another Nature journal. I note especially the following. (A) The inclusion of the new Figure 1(e) and the related text discussion in the present version of the paper is especially clarifying. Specifically, I agree with the authors, (1) that by considering B distortions perpendicular to the surface, and also a possible distortion in the S_m second layer, the 2×2 surface structure can have lowered rotational symmetry while keeping a square translational symmetry, and (2) that such kind of distortions could be large enough to produce the changes in the S_m wavefunctions that are implied by the authors' interpretation of their spectra. (B) The discussion of surface state counting from both a 1×1 and 2×2 viewpoint is a very good addition to the paper. (C) The added information on Fermi velocities and the Fermi contour areas is an important addition for making contact to other measurements. (D) The added Supplementary discussion of the possible role of the photoemission structure factor adds clarity to the authors' interpretation of their spectra.

I also think that the authors have responded very well to the comments that other reviewers made on the first version of the paper.

Based on the good response to reviewer comments, and on the general merit of the work, which I noted already in my previous report, and which is now further emphasized in the new version, I recommend acceptance of the paper in its present form.

Reviewer #3 (Remarks to the Author):

This is a high quality ARPES study that presents the first observation of a topological surface state Fermi surface with lower rotational symmetry than the bulk. My technical comments in the last review were rather minor, and have been well addressed by the resubmission. The revised manuscript has also provided a clearer discussion of the potential impact of these findings. This discussion is somewhat speculative, but I believe the novelty and quality of the work justify publication in Nature Communications.

(Reply to Reviewers)

Let us express our deepest acknowledgement to both reviewers. Thanks to their comments in the last rounds, we could improve our manuscript (MS) significantly. Now, we are delighted that they are supporting publication of our manuscript in Nature Communications.

REVIEWERS' COMMENTS

Reviewer #2 (Remarks to the Author):

The authors have responded substantively and satisfactorily to all the many comments that I made in a report on the version of the manuscript that was submitted to another Nature journal. I note especially the following. (A) The inclusion of the new Figure 1(e) and the related text discussion in the present version of the paper is especially clarifying. Specifically, I agree with the authors, (1) that by considering B distortions perpendicular to the surface, and also a possible distortion in the S_m second layer, the 2×2 surface structure can have lowered rotational symmetry while keeping a square translational symmetry, and (2) that such kind of distortions could be large enough to produce the changes in the S_m wavefunctions that are implied by the authors' interpretation of their spectra. (B) The discussion of surface state counting from both a 1×1 and 2×2 viewpoint is a very good addition to the paper. (C) The added information on Fermi velocities and the Fermi contour areas is an important addition for making contact to other measurements. (D) The added Supplementary discussion of the possible role of the photoemission structure factor adds clarity to the authors' interpretation of their spectra.

I also think that the authors have responded very well to the comments that other reviewers made on the first version of the paper.

Based on the good response to reviewer comments, and on the general merit of the work, which I noted already in my previous report, and which is now further emphasized in the new version, I recommend acceptance of the paper in its present form.

Reviewer #3 (Remarks to the Author):

This is a high quality ARPES study that presents the first observation of a topological surface state Fermi surface with lower rotational symmetry than the bulk. My technical

comments in the last review were rather minor, and have been well addressed by the resubmission. The revised manuscript has also provided a clearer discussion of the potential impact of these findings. This discussion is somewhat speculative, but I believe the novelty and quality of the work justify publication in Nature Communications.